# The Combination of Both Heat and Water Stresses May Worsen Botryosphaeria Dieback Symptoms in Grapevine

**DOI:** 10.3390/plants12040753

**Published:** 2023-02-08

**Authors:** Olivier Fernandez, Christelle Lemaître-Guillier, Aurélie Songy, Guillaume Robert-Siegwald, Marc-Henri Lebrun, Philippe Schmitt-Kopplin, Philippe Larignon, Marielle Adrian, Florence Fontaine

**Affiliations:** 1Unité Résistance Induite et Bioprotection des Plantes EA 4707, USC INRAE 1488, SFR Condorcet FR CNRS 3417, Université de Reims Champagne-Ardenne, 51100 Reims, France; 2Agroécologie, Institut Agro Dijon, CNRS, INRAE, Université Bourgogne Franche-Comté, 21000 Dijon, France; 3Independent Data Lab, 80937 Munich, Germany; 4Research Group Genomics of Plant-Pathogen Interactions, Research Unit Biologie et Gestion des Risques en Agriculture, UR 1290 BIOGER, Université Paris-Saclay, 78850 Thiverval-Grignon, France; 5Analytical BioGeoChemistry, Helmholtz Zentrum München, German Research Center for Environmental Health, 85764 Neuherberg, Germany; 6IFV Pôle Rhône-Méditerranée, 30230 Rodilhan, France

**Keywords:** transcriptomic, metabolomic, *Vitis vinifera*, abiotic stress, biotic stress, *Botryosphaeriaceae*

## Abstract

(1) Background: Grapevine trunk diseases (GTDs) have become a global threat to vineyards worldwide. These diseases share three main common features. First, they are caused by multiple pathogenic micro-organisms. Second, these pathogens often maintain a long latent phase, which makes any research in pathology and symptomatology challenging. Third, a consensus is raising to pinpoint combined abiotic stresses as a key factor contributing to disease symptom expression. (2) Methods: We analyzed the impact of combined abiotic stresses in grapevine cuttings artificially infected by two fungi involved in Botryosphaeria dieback (one of the major GTDs), *Neofusicoccum parvum* and *Diplodia seriata*. Fungal-infected and control plants were subjected to single or combined abiotic stresses (heat stress, drought stress or both). Disease intensity was monitored thanks to the measurement of necrosis area size. (3) Results and conclusions: Overall, our results suggest that combined stresses might have a stronger impact on disease intensity upon infection by the less virulent pathogen *Diplodia seriata*. This conclusion is discussed through the impact on plant physiology using metabolomic and transcriptomic analyses of leaves sampled for the different conditions.

## 1. Introduction

Grapevine trunk diseases (GTDs) have become a serious concern for winemakers worldwide, threatening the sustainability of vineyards in some countries [1,2,3]. Botryosphaeria dieback (BD), one of the most concerning GTDs [4], is caused by *Botryosphaeriaceae fungi*, including *Neofusicoccum parvum* and *Diplodia seriata*, two species commonly isolated in infected young grapevines and established vineyards [5]. Internal symptoms in woody tissues are brown wood streaking, necrotic lesions, discoloration in the outer xylem, and perennial cankers. External symptoms consist of bud necrosis or death, leaf intervein discoloration, dead arms, and shoot dieback [3,6,7,8], but they are notoriously inconsistent in incidence and prevalence from one year to another. One of the key features of GTDs is that responsible fungi can prevail in grapevine as endophytes for several years without any apparent symptoms [9,10,11].

It is now largely admitted that abiotic factors can trigger the expression of GTD symptoms (including BD) or at least favor their development [8,12,13]. Abiotic stresses, particularly heat and water stresses, alone or combined, affect both the plant (development, physiology, and production) and the pathogens (growth, distribution, and virulence), leading to changes in the outcome of the host–fungus interaction (for a review, see [14,15]). For example, positive correlations of BD foliar symptom expression with spring rainfall and rain episodes during the month prior to their appearance have been demonstrated [16]. Water stress, provoked by restricted watering regimes, may induce a higher colonization and more severe BD woody symptoms [17,18]. Likewise, high temperatures appeared to significantly enhance the virulence of *Botryosphaeriaceae* spp. [18,19]. Environmental stresses often occur simultaneously under field conditions [20,21]. The frequency of the occurrence of heat waves, extreme precipitation events, and droughts is likely to increase according to the recent consensus scenario of climate change (IPCC 2014), especially in grapevine growing areas [22,23]. Furthermore, the development of irrigation strategies aimed at ensuring grape quality while saving water resources sets vines on the edge of water stress and might make them more susceptible to heat stress [24,25,26,27]. Therefore, the study of the effect of simultaneous heat and drought stresses over the course of GTD symptom expression is of paramount importance for wine professionals.

The individual effects of abiotic and biotic stresses on plant physiology have been largely studied on grapevine. Transcriptional analyses from plants submitted to heat or drought stress alone have been performed [28,29,30,31,32]. For example, in [31] Rocheta et al. pointed out in the cultivar Aragonez (i) the activation of common signaling pathways and cellular responses by both stresses (ABA and ethylene signaling pathways), and (ii) specific response pathways to each single stress such as repression of a cysteine protease gene for drought and induction of heat-shock proteins (HSP) coding genes for heat stress.

Regarding the impact of GTDs on vine physiology, transcriptional and metabolomic changes have already been observed on established vines infected by GTDs [33,34,35,36,37]. An artificial infection by *N. parvum* leads to transcriptional changes in grapevine, particularly in the leaves [38,39].

However, it is widely admitted that the molecular responses, either transcriptomic or metabolomic, to a combination of stresses is unique and may not be extrapolated from the responses to the respective individual stress [21,40,41,42]. Unfortunately, to our knowledge, the effects of combined heat and drought have not yet been investigated in grapevine.

Another important feature is that the impact of abiotic stress on GTDs is often studied through symptom assessment and physiological measurements [15] and seldom via molecular studies [19,43,44]. Among these, in [44], Lima et al. studied the metabolic changes of the xylem sap after exposure to water stress of plants previously infected by spores of esca fungi, another major GTD. They reported an increase in the concentration of most metabolites under stress that may promote fungal growth. Colonization of the plant by GTD pathogens may also be favored under heat stress. Indeed, a transcriptomic analysis [19] performed by Paolinelli-Alfonso et al. revealed that the colonization of *Lasiodiplodia theobromae*, another fungus involved in BD, depends on fungal (i) ability to upregulate its genes encoding pectate lyase and xylosidase glycoside hydrolase for cell-wall degradation, and (ii) capacity to use, for its own metabolism, phenylpropanoid precursors further synthesized by the plant in response to heat stress. More recently, in [43], Galarneau et al. previously highlighted putative host-based molecular markers of infection for expression in plant drought-stressed or not, with or without infection by *N. parvum*. Lastly, there is simply no evaluation of the impact of the double heat and drought stress on BD infection, although this dual-stress condition is likely to become common.

In this paper, we precisely examined the impact of the combination of heat and water stresses on the grapevine interactions with *D. seriata* and *N. parvum*. First, we comment on the results on individual stress alone (heat, drought and BD) in order to assess the validity of our stress application protocols (Figure 1). Then, we analyze how combined abiotic stresses affect the plant responses to both fungi at the symptom expression, metabolic, and molecular levels, with a focus on the impact of double stress (heat and drought).

## 2. Results and Discussion

### 2.1. Measurement of Ds and Np Necrosis under Different Stress Combinations

The first question of this study was to assess whether abiotic stresses, single or combined, might interfere with symptom development following a controlled infection of grapevine plants with *N. parvum* and *D. seriata* (hereinafter Np and Ds, respectively). Necrosis length and area were both recorded 1 month after inoculation (Figure 2) on grapevine plants subjected to the experimental conditions summarized in Figure 1. The pathogens (Np and Ds) were successfully reisolated at timepoints A and B from all plants, regardless of the experimental conditions (data not shown).

Following variable transformation to ensure data normalization, two-way ANOVA with interaction (factor 1, infection with Np/Ds and factor 2, nature of the applied stress; Table 1) was performed on both necrosis parameters (necrosis length and area). For both necrosis parameters, the first factor (i.e., the nature of fungal pathogen used for the infection Np or Ds) was significantly different (Table 1). Necrosis caused by Np was longer and wider (Figure 2). This difference is in agreement with previous comparisons between infection with these two *Botryosphaeriaceae fungi*, where Np always seemed to be the more aggressive [7,45,46]. When assessing the second ANOVA factor, the effect of stresses appeared globally nonsignificant on both length and area of the necrosis (Table 1). However, depending on which fungus was used for the infection, the necrosis length was differently affected by the application of abiotic stresses. If Np infection appeared unaffected by the nature of the abiotic stress (Table 1), Ds necrosis length was significantly increased by the application of combined heat and water stress (Figure 2A; Table 1).

This observation suggests that the combination of abiotic stresses, at least in artificial infection conditions, might affect the outcome of BD, especially for *D. seriata*. Furthermore, the application of combined abiotic stresses appeared to increase the virulence of less aggressive pathogens such as *D. seriata*.

In the remainder of this section, we analyze transcriptomic and metabolic responses of the plant under our various experimental conditions (Figure 1) with a focus on whether these data support our phenotypic observations of the artificial infection.

### 2.2. Transcriptomic Analysis of Single Stress Responses

We first analyzed the response to single stresses (heat stress (hereinafter HS), water stress (hereinafter WS), and single GTD fungus) in order to assess the representativeness of the stress application protocol used herein (Figure 1) when compared to previous similar studies available on grapevine. All these comparisons were performed to the noninfected control + no abiotic stress (hereinafter NINAS) condition.

#### 2.2.1. Heat Stress Impact on Leaf Transcriptome

Similar to most previous studies on HS in grapevine, leaf samples were harvested within hours following the beginning of the temperature increase (1 h in [31]; 2 h in [47]; 3 h in [48] and the present study). At both timepoints (A and B), the transcriptome was affected by HS treatment. This is illustrated by the separation of the different conditions following PCA analysis (Figure 3A), as well as the number of differently expressed genes (DEGs) at both timepoints (Figure 4A).

DEGs were later classified using GO gene ontology. Among the most affected GO functional categories in the noninfected control + heat stress (hereinafter NIHS, Appendix A), the response to stimulus (GO:0050896), ion binding (GO:0005488), and intracellular anatomical structure (GO:0005622) were highly represented at timepoint A. At timepoint B, GO enriched categories concerned membrane parts or components (GO:0016020; GO:0016021; GO:00331224; GO:0031225). These GO categories are classically enriched following heat stress in grapevine [47], as well as following biotic stress [49].

Unsurprisingly, heat-shock protein-coding genes were among the most significant DEGs (Appendix A) at timepoint A. These chaperone-like proteins ensure correct folding of protein under stress and are classically upregulated in HS experiments [31,47,48]. In the meantime, cold-response protein-coding genes were downregulated, pointing toward a specific response to heat stress (Appendix A). Later in the experiment, heat-shock protein-coding genes were still upregulated but to a lesser extent (Appendix A). NIHS samples at timepoint B were also characterized by downregulation of several transcripts involved in pathogen-related protein and stilbene synthases (nine and 14, respectively, Appendix A). The latter was previously described in [31] and could be interpreted as a reduction of plant immunity following a prolonged HS. Lastly, several transcripts encoding putative starch related enzymes were found downregulated in NIHS leaf samples at timepoints A and B (Appendix A), as previously mentioned in [47].

#### 2.2.2. Water Stress Impact on Leaf Transcriptome

Noninfected control + water stress (hereinafter NIWS) leaf samples were harvested 8 days after the end of normal watering. Again, at both timepoints, PCA analysis clearly separated NIWS from NINAS leaf samples, even if WS-treated samples were less clearly grouped compared to HS treatment (Figure 3B). More DEGs were detected following WS compared to HS (Figure 4A), and GO functional categories were also differentially affected (Appendix A). Genes affected mainly belonged to the biological process category. Metabolic process (GO:0008152), especially the subcategory catabolic process (GO:0009056), catalytic activity (GO:0003824), and response to stimulus (GO:0050896) were over-represented at timepoint A (Appendix A). These three categories were also markedly affected in the study by [28].

Unlike at timepoint A, genes affected at timepoint B belonged to both the biological process and the molecular function categories (Appendix A). The binding category (GO:0005488) was significantly affected, along with catalytic activity (GO:0003824) and transport (GO:0006810). These GO categories are often affected by drought in several plants such as poplar [50], maize [51], and grapevine [52].

Overall, the response of grapevine leaves to individual abiotic stress appears to be in line with previous reports.

#### 2.2.3. Fungal Infection Impact on Leaf Transcriptome

Unlike both individual abiotic stress, fungal infection (*Diplodia seriata* inoculated + no abiotic stresses and *Neofusicoccum parvum* inoculated + no abiotic stresses leaf samples; hereinafter DsNAS and NpNAS, respectively) had a limited effect on grapevine transcriptional regulation. Using our threshold, it is worth mentioning that no genes were found transcriptionally affected in DsNAS samples at timepoint A, with only 20 affected at timepoint B (Figure 4B). These small modifications might account for the fact that changes in gene expressions were not recorded at the infection site but in distal leaves. NpNAS leaf samples were more affected. We found up to fivefold more DEGs (Figure 4B) and more GO categories (Appendix A) affected in this condition compared to DsNAS leaf samples, which could be related to the previously reported higher virulence of the fungi [45,46] and was confirmed by necrosis size in this study (Figure 2).

When examining DEGs at timepoint B, 11 DEGs appeared to be regulated in the same way in DsNAS and NpNAS leaf samples (Appendix A). Interestingly, three of these genes have already been associated with another biotic stress, i.e., in grapevine challenged by the pest insect *Tetranychus urticae* (VIT_201s0011g05180, VIT_212s0028g02990, and VIT_204s0008g05010, [53]). Two other DEGs were previously associated with grapevine stress response, VIT_202s0025g04460 under light stress following leaf removal [54] and VIT_209s0002g08510 following gibberellin treatment [55]. Lastly, one gene coding for a copper transporter (VIT_203s0110g00360) and two unknown proteins were found side to side on a cluster on chromosome 5 (VIT_205s0077g00770 and VIT_205s0077g00780).

### 2.3. Transcriptomic Analysis of Combined Stress Responses

The combination of HS and WS had a strong effect on the leaf sample transcriptome (noninfected control + water stress + heat stress samples; NIWSHS hereinafter) as illustrated by the PCA analysis (Figure 3C). Considering the number of DEGs, the effect of combined stresses was stronger than the effect of single stresses (Figure 4A).

A more detailed analysis of GO categories and top DEGs of the combined abiotic stresses revealed a complex interaction of their response. On one hand, the GO categories affected in NIWSHS showed strong similarities with the ones affected in NIWS, with several DEGs belonging to metabolic process (GO:0008152) and catalytic activity (GO:0003824, Appendix A). On the other hand, top DEGs detected in NIWSHS clearly resembled those detected in NIHS, especially an overexpression of heat-shock protein and a downregulation of several defense responses (26 DEGs related to stilbene synthase and five related to pathogens were downregulated, Appendix A). In a very recent study [56], Ju et al. also observed the prevalence of HSP genes under combined HS and WS. They also mentioned the importance of the involvement of spliceosome-related genes in their combined stress experiment which we did not detect in our samples. This might be related to the difference in their experiment which was carried out at a higher temperature (45 °C).

### 2.4. Metabolomic Analysis of Single Stress Responses

Along with transcriptomic analysis, we performed FT-ICR-MS analysis to study the impact of abiotic or/and biotic stresses echoed on the leaf metabolome. The analysis of all samples generated 6657 raw *m*/*z*, among which 5728 formulas were validated. Among the latter, 1152 *m*/*z* could be identified after queries in MassTrix (Appendix A). Data from all NI samples were isolated at timepoint A and B subsets. Statistical analysis led to the identification of 199 and 212 significant *m*/*z* discriminants for NINAS, NIWS, NIHS, and NIWSHS conditions at timepoints A and B, respectively (Appendix A). A comparison was then made between each single stress and the NINAS control (Figure 5A).

#### 2.4.1. Water Stress Impact on Leaf Metabolome

WS provoked a limited down-accumulation of metabolites in leaves at timepoint A (7 *m*/*z*). This down-accumulation was even more pronounced at timepoint B, with 95 *m*/*z* significantly down-accumulated (Figure 5A), whereas 14 *m*/*z* were up-accumulated. From the latter, only 18% could be annotated and categorized as 7% “lipids”, 5% “phytochemicals”, and 4% “carbohydrates” (not shown).

#### 2.4.2. Heat Stress Impact on Leaf Metabolome

The metabolic alteration profile was different for HS. Heat stress induced both up- and down-accumulation, in a time-dependent manner (Figure 5A). HS indeed induced the up-accumulation of 23 and 67 *m*/*z* at timepoints A and B, respectively. Among the latter, 21% could be annotated, including 15% “phytochemicals”, 3% “lipids”, and 3% “carbohydrates” (not shown). The number of down-accumulated *m*/*z* remained constant (82 and 83 *m*/*z* for timepoints A and B, respectively). From those down-accumulated compounds, 7% (including 4% “phytochemicals”) and 18% (including 7% “lipids”, 5% “phytochemicals”, and 4% “carbohydrates”) were annotated at timepoints A and B, respectively (not shown).

#### 2.4.3. Fungal Infection Impact on Leaf Metabolome

To only evaluate biotic stress effects on the leaf metabolome, NpNAS and DsNAS samples were compared to NINAS ones (Figure 6). Both fungi induced a down-accumulation of leaf metabolites, with a stronger and faster effect of *N. parvum*. In NpNAS samples, 33 and 224 *m*/*z* were down-accumulated at timepoints A and B, respectively. In DsNAS samples, there was no accumulated *m*/*z* at time A, and 139 *m*/*z* accumulated at time B. Functional categorization of all timepoint B annotated compounds revealed mainly “lipids”, “carbohydrates”, and, to a lesser extent, “phytochemicals” (29%, 21%, and 17%, respectively, for Np and 25%, 23%, and 17%, respectively, for Ds; Appendix A). Interestingly, a large number of regulated *m*/*z* (105) were common for NpNAS and DsNAS samples (Figure 7). Only 15 (14%) of them could be annotated, including 9% “phytochemicals” and 2% “carbohydrates” (Appendix A).

As infections were made at the level of stem internodes, metabolic changes detected in leaves clearly demonstrate that infection induces distant plant responses. Since they appear more extensive than transcription regulations described earlier, they might be at least partly related to metabolites flux via the vascular systems. In addition, these changes might be triggered by hormone signaling (jasmonic or salicylic acid, e.g., [57]), sugar signaling (e.g., [58]), or fungal toxins transfer via the xylem sap [9,59]. As for the transcriptomic analysis, the higher number of down-accumulated *m*/*z* for Np challenge might account for the higher virulence of this fungus [45,46].

Some down-accumulated *m*/*z* belong to the “lipids” and “carbohydrates” categories, which appears to be typical in GTDs. In a study of Esca disease, in [36], Magnin-Robert et al. also reported that lipid and carbohydrate metabolisms were strongly affected prior to apoplexy appearance. More recently, in [60], Labois et al. found a similar reduction in lipids and carbohydrates in woody tissue. In [34], Lemaitre-Guillier et al. highlighted the accumulation of lipids in the apparently healthy wood adjacent to the typical brown stripe of vines infected by Botryosphaeria dieback.

### 2.5. Metabolomic Analysis of Combined Stress Responses

As for the transcriptome response, NIWSHS appeared to be the most affected modality when considering metabolome analysis at timepoint A, with a high and similar number of up- and down-accumulated *m*/*z* at timepoint A (83 and 88 *m*/*z* were up- and down-accumulated, respectively, Figure 5A). However, the number of regulated *m*/*z* decreased at timepoint B, faster than HS stress alone, pointing a possible recovery that was not observed for the transcriptomic response. From the 83 accumulated *m*/*z* at timepoint A, only 7% were annotated, including 5% “phytochemicals”. From the 88 down-accumulated ones at timepoint A, only 10% were annotated, including 5% “phytochemicals”, 3% “lipids”, and 2% “carbohydrates”. At timepoint B, 25 and 43 *m*/*z* remained up- and down-accumulated, with 16% (including 8% “peptides”, 4% “carbohydrates”, and 4% “amino sugars”) and 20% annotated compounds (including 9% “phytochemicals”, 5% “carbohydrates”, 2% “lipids”, and 2% “peptides”), respectively (not shown).

Moreover, the same *m*/*z* were not involved at both times, since only 21 *m*/*z* were in common, while 156 *m*/*z* were specific to timepoint A and 173 *m*/*z* were specific to timepoint B, in terms of both up- and down-accumulation (Figure 5B). The corresponding annotated compounds mostly belong to the “phytochemicals” category (Appendix A).

Abiotic stresses are known to impact vine development and physiology (for a review, see [15]). In the present experiment, the combined WS and HS had a greater impact on these functions than a single stress, with a reduction in photosynthesis and growth of cuttings. According to these results, we observed the highest changes in the leaf metabolome in WSHS samples. HS and WS are known to affect the metabolic profile in grapevine, but most studies have been conducted on berries and have focused on an individual stress, making comparison difficult. In a study focusing on 119 compounds from primary metabolism, i.e., polyphenols and volatile compounds in leaves of Pinot Noir [61], Griesser et al. found that 60 compounds could discriminate control from drought-stressed samples. When performing leaf metabolic profiling of drought stress in Shiraz and Cabernet sauvignon plants grown in a greenhouse [62], Hochberg et al. reported late changes (18 to 34 days post the onset of water deficit) of the leaf metabolome. They observed a global upregulation of amino acids at 34 dpt (days post treatment) and a downregulation of organic acids, carbohydrates, and secondary metabolites from 18 dpt. This was not observed in the present study possibly since analyses were performed earlier (8 and 11 dpt), prior to these metabolic adjustments. In this respect, we observed a higher impact of water stress at timepoint B than at timepoint A. In [63], Cramer et al. also reported alteration in the relative abundance of few organic acids, amino acids (targeted analysis), or sugars in grapevine leaves 16 days after the onset of a water set. Only three were accumulated in our present analysis (malate, proline, and glucose).

These results are consistent with previous studies which reported that the plant responses arising from multiple stresses are unique and cannot be deduced by extrapolating from the responses to each stress individually (for a review, see [15]). Such a multi-stress context has to be further studied to better drive crop production, especially in the context of climate evolution with more frequent temperature (heat or cold) and water stresses (IPCC 2014). It also has to be considered for plant–microbe interactions [64].

### 2.6. Molecular and Metabolic Response when Combining Infection an Abiotic Stresses

There is now a large consensus regarding the connection between GTD emergence in vineyards and the appearance of abiotic stresses [15]. A plausible explanation might be that abiotic stresses favor or accelerate the transition from the latent to pathogen phase of the fungi [15]. Another possibility is that abiotic conditions might weaken the plant response to biotic factors, which prompted our study. In this article, our phenotypic analysis of necrosis development is clearly in favor of the latter possibility (Figure 2). Elements from the subsequent analysis of the leaf transcriptome and metabolome of our samples appear to also concur with it.

#### 2.6.1. Analysis of Leaf Transcriptome following Combined Biotic and Abiotic Interaction

Abiotic stresses, single or combined, appeared to have a stronger impact on Ds-infected samples compared to Np-infected ones. A maximum of 47 genes were differentially affected in Np-infected samples (NpWS timepoint B, Figure 4C), whereas up to 126 were affected in Ds-infected samples (*Diplodia seriata* inoculated + heat stress (hereinafter DsHS) timepoint A, Figure 4C). This is consistent with our view of how both pathogen infections might interact with abiotic stresses.

Cuttings appeared more damaged following Np infection (Figure 1), and their leaf transcriptome was found strongly responsive to the pathogen (Figure 4B). Following Np infection, the plant is either less receptive to additional abiotic stress due to infection by a more virulent pathogen, or some of plant responses between biotic and abiotic stress are cumulative.

Following Ds infection, the number of DEGs was consistently higher when combining the action of an abiotic stress to the infection compared to Np (Figure 4C,D). Following HS and WS, this was especially observed at timepoint B (300 and 106 DEGs, respectively), while this was the case at timepoint A for the double abiotic stress (WSHS, 94 DEGs).

Overall, we believe this to be in agreement with the existing literature on Np fungus, which is clearly more aggressive than most of the other *Botryosphaeriaceae* [45,65,66]. Following this reasoning, Ds, being less aggressive alone, “needs” an interaction with given abiotic conditions in order to have a strong impact on the plant molecular response. This could explain the emergence of BD in field in relation to extreme weather events. We can speculate that, if a large number of less aggressive pathogens such as Ds are present in latent form endophytically in the plant, their aggressive behavior might be triggered by individual or combined abiotic stresses.

Interestingly, several genes involved in plant defense response were upregulated following *Diplodia seriata* inoculated + water stress + heat stress (hereinafter DsWSHS) and Np alone. A possible explanation is that the perception of Ds is only effective and activates the defense response when another signal such as abiotic stress is present.

At the molecular level, the impact of HS appeared stronger on Ds samples compared to either WS or even WSHS (Figure 4C). This does not follow the pattern of necrosis enlargement that was observed in Figure 2 (the impact of WSHS was clearly higher on the necrosis area). One possible explanation is that the impact of the double stress is not measurable at the transcriptomic level but rather later in the metabolome (see next section).

#### 2.6.2. Analysis of Leaf Metabolome following Biotic and Abiotic Interactions

Overall, a global metabolite downregulation was observed when comparing samples of infected vines submitted to abiotic stress, single or combined (*Neofusicoccum parvum* inoculated + water stress, hereinafter NpWS; *Neofusicoccum parvum* inoculated + water stress, hereinafter NpHS; *Neofusicoccum parvum* inoculated + water stress + heat stress, hereinafter NpWSHS; DsHS; DsHS; DsWSHS), to the corresponding noninfected NI ones (Figure 7). Appendix A indicates the number of regulated *m*/*z* for which a validated raw formula could be obtained. WS had the lower impact on the leaf metabolome except for Np-infected samples at timepoint B (NpWS, 157 *m*/*z*: with 8 and 149 up- and down-accumulated *m*/*z* (Figure 7). Among them, 33 were annotated (Appendix A). Compared to WS, HS had a higher impact on the leaf metabolome of infected samples, especially at timepoint A. At this time, 13 and 570 *m*/*z* were up- and down-accumulated, respectively, for Np samples while 15 and 523 *m*/*z* were up- and down-accumulated for Ds samples (Figure 7). Among those 583 and 538 *m*/*z*, 131 and 128 could be annotated, respectively (41 and 80 top ones in Appendix A). At timepoint B, only 1 *m*/*z* remained up-accumulated for Np samples, and 107 remained down-accumulated for Ds ones. Among the most down-accumulated compounds that could be annotated in NpHS samples, we observed glucose, the stilbenes polydatin and piceatannol, a jasmonate derivative, abscisic acid, and derivatives of the phenolics naringenin, catechin, and quercetin. The decreases in DsHS samples also encompassed glucose, the stilbenes piceid and piceatannol, resveratrol, astringin, and derivatives of naringenin, catechin, and quercetin (Appendix A). More than 80% of the downregulated ones (i.e., 486 *m*/*z*) were common to NpHS and DsHS samples (Figure 7). From those common compounds, 22% (i.e., 106 ones) could be annotated, including 11% “phytochemicals”, 6% “carbohydrates”, and 3% “lipids” (Appendix A).

A stronger regulation was observed in response to the double abiotic stress WSHS, although it was limited to timepoint A. As observed in Figure 7, 882 *m*/*z* (37 *m*/*z* up- and 845 *m*/*z* down-accumulated) in NpWSHS samples and 802 *m*/*z* (3 up- and 799 down-accumulated) in DsWSHS samples were recorded (Figure 7). Among them, 187 and 174 were annotated, respectively (see list of the 39 and 51 top ones in Appendix A). As in response to heat stress, more than 80% of the down-accumulated ones were common to NpWSHS and DsWSHS samples (761 *m*/*z*) (Figure 7). Among those common compounds, 22% (i.e., 165 ones) could be annotated, including 11% “phytochemicals”, 5% “carbohydrates”, and 3% “lipids” (Appendix A). Among the most down-accumulated compounds that could be annotated in WSHS samples, we observed abscisic acid and gibberellins. In addition, there were glucose, the stilbene polydatine, a derivative of catechin, and a derivative of jasmonic acid, together with free and conjugated abscisic acid (Appendix A).

Altogether, our results suggest that drought had a lower overall metabolic impact on infected grapevines compared to heat stress, and even more so compared to combined WSHS. The impact of HS and WSHS on the leaf metabolome of infected vines was severe and occurred faster than in uninfected vines, suggesting a strong and rapid plant response. However, the “recovery” to a basal metabolome status occurred sooner, in comparison to water stress. In uninfected grapevines, evolution toward such a “recovery” status was only observed for the combined stress (Figure 7). Furthermore, in terms of the number of affected compounds, the impact of the stress was more severe for leaves of infected plants, highlighting a specific plant response to combined biotic and abiotic stresses.

Abiotic stresses indeed have effects on micro-organisms (development and virulence) and on plant physiology and, in connection, with the performance of its immune system [67]. Abiotic stresses can, therefore, modify the outcome of plant–pathogen interactions. More generally, environmental factors have been recognized as modulators of plant immunity [67] and important predisposing factors for disease expression. This is especially true for GTDs [8,12,13]. Water stress indeed induces a higher vine colonization by fungi responsible for Botryosphaeria dieback and more severe symptoms [17,18]. The virulence of *Botryosphaeriaceae* spp. is increased by high temperatures [18,19]. The present study highlighted a decrease in stilbenes known as grapevine phytoalexins [68,69], and phenolics derivatives in leaves of heat-stressed Np- and Ds-infected plants at timepoint A, compared to non-inoculated ones. These results show that the vine response to heat stress involves the phenylalanine polymalonate pathway, which is shut down in biotically stressed vines. This flaw in the vine’s defense response is likely to favor the development of pathogens. Stilbenes and other phenolics are indeed active against several grapevine fungal pathogens, including those of the *Botryosphaeriaceae* family. As suggested by glucose down-accumulation, mobilization of energy to fuel defenses might also be deficient. When combined, water and heat stress differently affected the leaf metabolome in infected plants. Even if glucose can also be down-accumulated (for Np but not for Ds infection), the phenylalanine polymalonate pathway is quite no longer affected, and there is a down-accumulation of phytohormones abscisic acid, gibberellins, free and conjugated abscisic acid, and a derivative of jasmonic acid. This suggests a crosstalk between signaling hormones that can also modulate the outcome of pathogen infection [70]. Studies of multiple abiotic and biotic stresses have shown that these interactions can be both positive and negative depending on the specific stresses/pathogen interaction [20,21]. As shown, the importance of studying multi-stress contexts specifically cannot be overstated, since responses arising from multi-stresses are unique and cannot be deducted by extrapolating from each individual stress response [41,71].

### 2.7. Conclusions

Two key general conclusions might be drawn for our work. First, as we previously exposed in a recent review [15], abiotic stress appears to play a key role in triggering or at least aggravating GTD emergence. Impact of the infection on plant transcriptome and metabolome was always more pronounced when the infection was combined with an abiotic stress and even more when both heat and water stresses were combined.

Second, it seems that the impact of combined abiotic stress might raise the plant sensitivity to less virulent pathogens such as *D. seriata* compared to *N. parvum* herein. These findings could open a new direction in the management of GTDs. Indeed, if temperature cannot be manipulated in the vineyard, a fine piloting of irrigation could limit the risk of infections when a heat wave rages in a given field. Such fine-tuning of irrigation may be more important in the future since an increasing number of vineyards would undergo drought in the coming climate change conditions and more irrigation, as initiated by [72] in a GTD situation.

## 3. Materials and Methods

### 3.1. Experimental Design

#### 3.1.1. Plant Material and Growth Conditions

From mother grapevines certified free of GLFV (grapevine fanleaf virus), ArMV (Arabic mosaic virus), and GLRaV1, two and three (grapevine leafroll-associated virus), three-node wood vegetative cuttings of *Vitis vinifera* cv. Ugni blanc clone 384 were produced by the Institut Français de la Vigne et du Vin, Pôle Sud Ouest and then transferred to the University of Reims Champagne-Ardenne. At the stage of seven spread leaves, they were transplanted into 2 L pots containing 1.8 kg (dry weight) of a mixture of 70% of sand and 30% of peat moss. The field capacity of the soil was 21.1%. All the cuttings were weighed, and that value was used as a calibrator for future water adjustment. Cuttings were then watered in excess and left for 3 weeks in the greenhouse under standard conditions for grapevine growth [73]. The height of shoots was limited to 10 leaves by removing the apex when the 10th leaf had spread and buds at the axil of leaves were removed. During these 3 weeks, each plant was fertilized once a week with 30 mL of a modified Lesaint solution. After inoculation, cuttings were placed for the 2 month experiment in a growth chamber at 25 °C in the day and 15 °C in the night, with a relative humidity of 50% and a photoperiod of 16/8 h (PAR ≈ 200 µmol·m^−2^·s^−1^). The watering was adjusted to 100% of the soil field capacity. In order to attenuate the fluctuation in soil water content, the potted plants were weighed daily, and the quantity of water was adjusted to reach or maintain the desired field capacity.

#### 3.1.2. Artificial Fungal Inoculation

Fungal inoculation was performed 3 weeks after the transplanting when cuttings were at the developmental stage of 10 spread leaves. The pathogen, either *Diplodia seriata* strain 98.1 (Ds) or *Neofusicoccum parvum* strain Bourgogne (Np), was artificially inoculated in the wounded third internode of the green stem (from the base of the stem), according to [45,74]. A plug of mycelium from a 7 day old culture was used, and the wound was of 5 mm length and width and 1 mm depth. A potato dextrose agar plug was used as a control (NI) on similarly wounded stems.

#### 3.1.3. Abiotic Stress Treatments

Water stress (WS), heat stress (HS), or a combination of both (WSHS) was applied. Control plants (NAS) were kept at 25 °C in the day and 15 °C in the night and watered at 100% field capacity throughout the experiment. For WS, drought was progressively imposed from 8 days post inoculation (dpi; Figure 1). Water supply was progressively reduced until reaching 25% of field capacity in 8 days as follows: 90% of field capacity on day 1 of the WS, 80% on day 2, 70% on day 3, 60% on day 4, 50% on day 5, 40% on day 6, 30% on day 7, and 25% on day 8. Then, the field capacity of the WS plants was maintained at 25% on days 9 and 10. Finally, all the plants were watered again at 100% of field capacity until the end of the experiment. Meanwhile, the water supply of the other treatments was maintained at 100% of the field capacity.

HS was implemented on the eighth day of WS, i.e., at 25% of field capacity (Figure 1). HS plants were placed for 3 days at a temperature of 35 °C in the day and 18 °C in the night with similar light and relative humidity. On the day of the recovery of the WS, all plants were transferred into the growth chamber regulated at 25 °C in the day and 15 °C in the night. Until the end of the experiment, the plants were kept in the standard growth conditions.

To summarize, the experimental design consisted of the following groups: (i) noninfected control + no abiotic stress (NINAS); (ii) noninfected control + water stress (NIWS); (iii) noninfected control + heat stress (NIHS); (iv) noninfected control + water stress + heat stress (NIWSHS); (v) *Diplodia seriata* inoculated + no abiotic stresses (DsNAS); (vi) *Diplodia seriata* inoculated + water stress (DsWS); (vii) *Diplodia seriata* inoculated + heat stress (DsHS); (viii) *Diplodia seriata* inoculated + water stress + heat stress (DsWSHS); (ix) *Neofusicoccum parvum* inoculated + no abiotic stresses (NpNAS); (x) *Neofusicoccum parvum* inoculated + water stress (NpWS); (xi) *Neofusicoccum parvum* inoculated + heat stress (NpHS); (xii) *Neofusicoccum parvum* inoculated + water stress + heat stress (NpWSHS). Each condition contained a total of 12 plant replicates randomly selected.

### 3.2. Sample Collection

Sampling was performed at the beginning of HS (timepoint A) and the end of HS (timepoint B; Figure 1). Timepoint A corresponds to 1 h after 35 °C was established in the growth chamber. Timepoint B corresponds to the end of the third day of HS (2 h before the end of the day, after 14 h of light). For all molecular analyses, three leaves above the inoculation point of the pathogen were collected on three plants per condition, immediately frozen in liquid nitrogen, and stored at −80 °C.

### 3.3. Phytopathogen Re-Isolation and Necrosis Measurement

At each sampling timepoint (A and B), re-isolations were made to verify the presence of the inoculated pathogen (*D. seriata* or *N. parvum*) and its absence in the wounded controls (NI), according to the protocol of [75]. Then, 2 months post inoculation, the length and area of external cankers and internal necrosis developed on the green stem were measured using the software ImageJ (v1.48). For each timepoint, three plants per condition were used.

### 3.4. Transcriptomic Analysis

#### 3.4.1. RNA Extraction, cDNA Library Construction, and Illumina RNA Sequencing

Total RNA was isolated from 3 × 50 mg of leaf powder using the PureLink Plant RNA Purification Reagent (Invitrogen, Cergy-Pontoise, France). The manufacturer’s protocol was followed until the phase of separation with chloroform/isoamylic alcohol (24:1). Then, one volume of ethanol 70% was added to the aqueous solution obtained. This new solution containing the total RNA was purified using the NucleoSpin RNA kit (Macherey-Nagel, Düren, Germany) according to the manufacturer’s instructions. However, the incubation time for the rDNase was shortened to 8 min, and the final elution was only made in 30 µL of RNase-free water. For the purification step, the three technical replicates of the same sample were reunified on one column to maximize the final total RNA quantity. RNA was checked for integrity making an electrophoresis gel, and then quantified and quality-checked with the Nanodrop (Ratio A260/A280 = 1.8–2.2). The quality was finally analyzed with Experion RNA StdSens Chips (Bio-Rad) to verify intact ribosomal bands (RNA Quality Indicator (RQI) values ≥6). RNA sequencing was performed by Genewiz (South Plainfield, New Jersey, USA) using the Illumina HiSeq sequencing system with the 2 × 150 bp read length configuration and three lanes. The cDNA was generated by the company from polyA-purified total RNA.

#### 3.4.2. Read Mapping, Assembly, and Differential-Expression Analysis

Raw Fastq files were quality-controlled with FastQC (v0.11.7 [76]), and then trimmed for adapters using cutadapt (v1.16 [77]). For this analysis, the genome used was the 12X version of PN40024 (Pinot Noir [78]) and the annotation used was the “V2.1 CRIBI” [79]. A file containing all isoforms sequences described in the annotation V2.1 CRIBI was generated by gffread (v0.9.12; http://ccb.jhu.edu/software/stringtie/gff.shtml, accessed on 1 March 2018). For all samples, the number of reads mapped to each isoform was counted using Salmon (v0.9.1 [80]). The count per isoform was turned into a count per gene using the package tximport (v1.8.0 [81]) of the software R (v3.5.0). Then, the differentially expressed genes (DEGs) between two groups of samples were detected using the R package EdgeR (v3.22.2 [82]), thanks to the script “run_DE_analysis” of the Trinity suite [83]. Lastly, the DEGs were filtered with the following parameters: an absolute value of log_2_ of the fold change (FC) between two groups superior or equal to 1 (log_2_(FC) ≥ 1) and an adjusted *p*-value (P) associated with a comparison inferior or equal to 0.05 (FDR ≤ 0.05).

To study the effect of abiotic stresses, alone or in combination, on the NI plants, we used as control all the NI NAS plants at both timepoints A and B (six plants), with the kinetic effect on this control group being considered negligible. In the same way, the kinetic effect on the NIWS plants was not studied, and the general effect of water stress was analyzed comparing all the NIWS plants at timepoints A and B (six plants) to the control group (NINAS). For the analysis of the heat stress effect, alone or in combination, the kinetics was considered. For instance, to study the effect of heat stress at a specific timepoint, the NIHS plants at this timepoint (three plants) were compared to the control group (six plants). Then, to investigate the impact of infection in each abiotic stress condition, several control groups were used. The infected plants with a certain abiotic stress condition at a specific timepoint (three plants) were compared to the NI control with the same abiotic stress condition at the same timepoint (three plants). Lastly, a comparison study was made to compare *N. parvum* and *D. seriata* infection. Ds plants with a certain abiotic stress condition and at a specific timepoint (3 plants) were compared to Np plants with the same abiotic stress condition and at the same timepoint (three plants).

#### 3.4.3. Gene Ontology Analysis

In order to characterize the DEGs in drought treatment, heat treatment, or the combination of both, GO-based enrichment tests were carried out using the software agriGO (v2.0) (available at http://systemsbiology.cpolar.cn/agriGOv2/, accessed on 1 March 2018). Enriched GO terms were detected using singular enrichment analysis (SEA) with the *Vitis vinifera* “Gramene Release 50” as GO annotation reference. The significance of over-represented terms in the three categories (biological process, cellular component, and molecular function) were assessed by Fisher’s exact test, adjusted by the Hochberg (FDR) method. The significance level and the minimum number of mapping entries were set at 0.05 and 5, respectively [84,85].

#### 3.4.4. Illumina RNA-Seq Results Validation by qRT-PCR

The analysis was performed on the same samples as the Illumina sequencing. In total, 150 ng of total RNA was reversed-transcribed, using Verso cDNA synthesis kit (Thermo Fischer Scientific, Surrey, U.K.) according to the manufacturer’s protocol. PCR conditions were as described in [86]. The expression of 10 targeted genes selected from transcriptomic analysis was tracked by quantitative reverse transcription polymerase chain reaction (qRT-PCR) using the primers indicated in Appendix A. Reactions were carried out in a real-time PCR detector Chromo 4 apparatus (Bio-Rad) using the following thermal profile: 15 s at 95 °C (denaturation) and 1 min at 60 °C (annealing/extension) for 40 cycles. The efficiency of the primer sets was estimated by performing real-time PCR on several dilutions. PCR reactions were performed in duplicate on three biological replicates per condition. The data were analyzed using the Bio-Rad CFX Manager software (v3.0), and the relative gene expression levels were determined by the method of [87], with EF1-α and 39SRP as internal reference genes. The results were log_2_-transformed and correspond to the means ± standard error of the biological triplicate in most cases, except for the study of water stress effect in which six biological replicates were considered. Contrasting with the Illumina RNA-Seq results, to study the effect of abiotic stresses, the results for timepoints A and B were kept separated and the controls used for the qRT-PCR analysis were the non-stressed plants (three plants) of the same kinetic time. To investigate the infection effect, the controls were similar to the ones used previously in the RNA-Seq analysis. The analyzed genes were considered significantly up- or downregulated when their relative expressions were >1 or <−1, respectively. This validation procedure is discussed in the Appendix A.

### 3.5. Metabolomic Analysis

#### 3.5.1. FT-ICR-MS Analysis

Prior to analysis, 15 mg of each sample was prepared in methanol (LC–MS grade, Fluka Analytical; Sigma-Aldrich, St. Louis, MO, USA) as previously described [88]. Ultrahigh-resolution mass spectra were acquired using an ion cyclotron resonance fourier-transform mass spectrometer (FTICR-MS) (solariX, Bruker Daltonics GmbH, Bremen, Germany) equipped with a 12 T superconducting magnet (Magnex Scientific Inc., Yarnton, GB, USA) and an APOLO II ESI source (Bruker Daltonics GmbH, Bremen, Germany) operated in the negative ionization mode. Samples were introduced into the micro electrospray source at a flow rate of 120 μL·h^−1^. Spectra were acquired with a time domain of 4 megawords over a mass range of 100 to 1000, and 300 scans were accumulated per sample.

#### 3.5.2. Data Analysis

Spectra were externally calibrated on clusters of arginine (10 mg·L^−1^ in methanol). Further internal calibration was performed for each sample by using a list of ubiquitous fatty acids and recurrent wine compounds, allowing mass accuracies of 0.1 ppm [89]. The *m*/*z* peaks with a signal-to-noise ratio (S/N) of 4 and higher were exported to peak lists.

All *m*/*z* detected were given an identification number (ID) in order to simplify further data processing. Several step procedures were then applied on the data lists (*m*/*z*). First, Perseus software (https://maxquant.net/perseus/) selected *m*/*z* when seen at least three times in datasets. These *m*/*z* coupled to their respective peak intensities were submitted to statistical analysis with Perseus software, which performed hierarchical clustering and multivariate ANOVA analyses between samples groups under a *p* < 0.05 to get significant *m*/*z*. All significant *m*/*z* were then submitted to compound identification using MassTrix software (http://masstrix3.helmholtz-muenchen.de/masstrix3/ accessed on 1 June 2018), which proceeds to search against KEGG, HMDB and LipidMaps databases (at 1 ppm tolerance value), on the *Vitis vinifera* organism. Compounds identified as isotopes (C13, N15, O18, etc.) were kept in the list of interest. In parallel, *m*/*z* were subjected to the NetCalc algorithm and an in-house software tool to obtain elemental chemical formulae [90] validated by van Krevelen diagram calculation, as previously described [34]. The *m*/*z* with reliable row formulas were manually classified into predicted functional categories (lipids, peptides, amino sugars, carbohydrates, nucleotides, phytochemicals, and NM when not matching or drug classified) according to the KEGG annotations. The category “phytochemicals” corresponds to secondary metabolites. Raw formula agreements on identified compounds between NetCalc and MassTrix were verified and revised when necessary. Indeed, wrongly calculated formulas encompassed isotopes and other elements (Br, Cl, F, Ca, etc.) containing molecules. These compounds were considered and specified in tables but were not revised or reclassified and left as the NM functional category. Top disease-specific compounds relative to control were sorted among the identified compounds by varying *t*-test parameters such as S0 (diseased:control fold parameters) and FDR to more stringent values in Perseus software.

## Figures and Tables

**Figure 1 plants-12-00753-f001:**
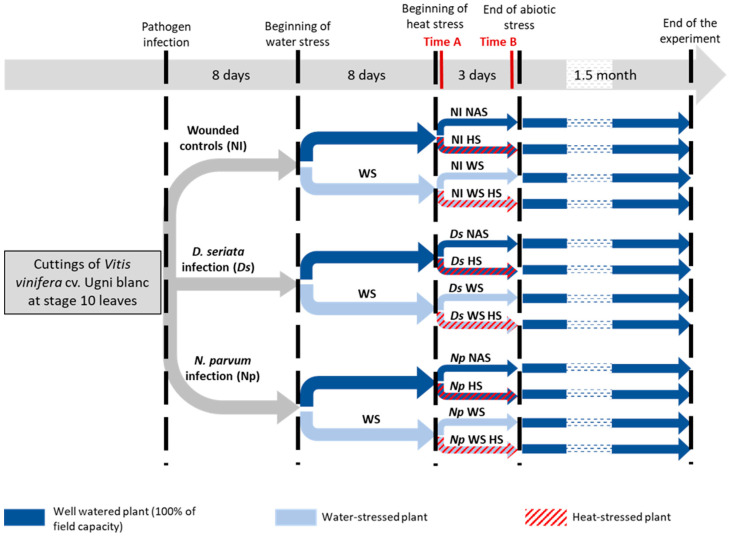
Summary of the main steps of the experiment. Both sampling times are represented in red (time A and B). Water stress was progressively imposed to reach 25% of field capacity during heat stress. Heat stress was set at 35 °C in the day and 18 °C in the night comparatively to the non-abiotic stress control placed at 25 °C in the day and 15 °C in the night. Stress condition abbreviations: (i) noninfected control + no abiotic stress (NINAS); (ii) noninfected control + water stress (NIWS); (iii) noninfected control + heat stress (NIHS); (iv) noninfected control + water stress + heat stress (NIWSHS); (v) *Diplodia seriata* inoculated + no abiotic stresses (DsNAS); (vi) *Diplodia seriata* inoculated + water stress (DsWS); (vii) *Diplodia seriata* inoculated + heat stress (DsHS); (viii) *Diplodia seriata* inoculated + water stress + heat stress (DsWSHS); (ix) *Neofusicoccum parvum* inoculated + no abiotic stresses (NpNAS); (x) *Neofusicoccum parvum* inoculated + water stress (NpWS); (xi) *Neofusicoccum parvum* inoculated + heat stress (NpHS); (xii) *Neofusicoccum parvum* inoculated + water stress + heat stress (NpWSHS).

**Figure 2 plants-12-00753-f002:**
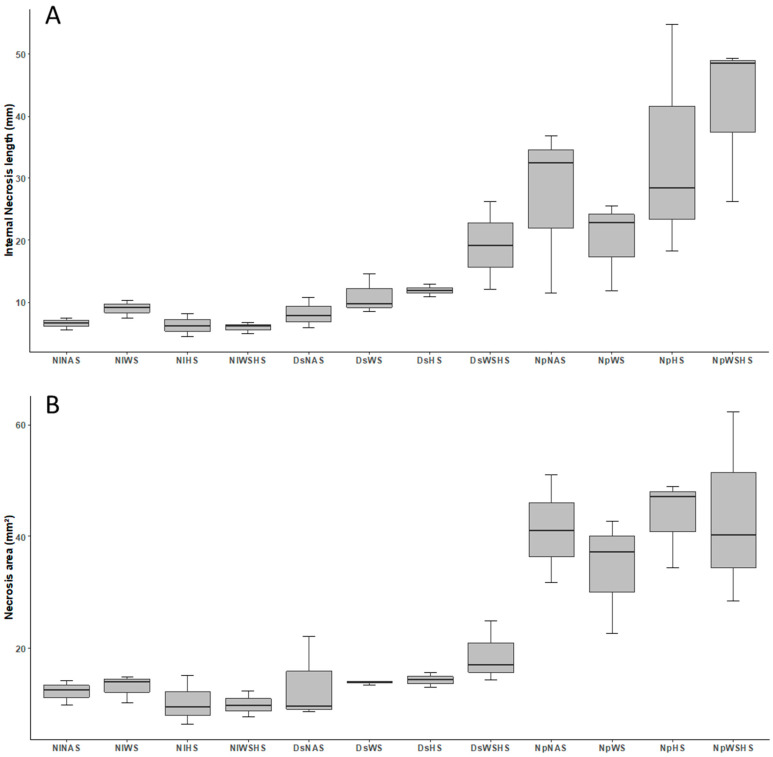
Internal necrosis length (**A**) and necrosis area (**B**) measured on grapevine stems following the various treatments described in Figure 1. Samples subjected or not to abiotic stress are represented in this figure. The statistical information displayed on the graph denotes the significant interaction with stress for given infection conditions (see Table 1 for two-way ANOVA detailed results). Stress condition abbreviations: (i) noninfected control + no abiotic stress (NINAS); (ii) noninfected control + water stress (NIWS); (iii) noninfected control + heat stress (NIHS); (iv) noninfected control + water stress + heat stress (NIWSHS); (v) *Diplodia seriata* inoculated + no abiotic stresses (DsNAS); (vi) *Diplodia seriata* inoculated + water stress (DsWS); (vii) *Diplodia seriata* inoculated + heat stress (DsHS); (viii) *Diplodia seriata* inoculated + water stress + heat stress (DsWSHS); (ix) *Neofusicoccum parvum* inoculated + no abiotic stresses (NpNAS); (x) *Neofusicoccum parvum* inoculated + water stress (NpWS); (xi) *Neofusicoccum parvum* inoculated + heat stress (NpHS); (xii) *Neofusicoccum parvum* inoculated + water stress + heat stress (NpWSHS).

**Figure 3 plants-12-00753-f003:**
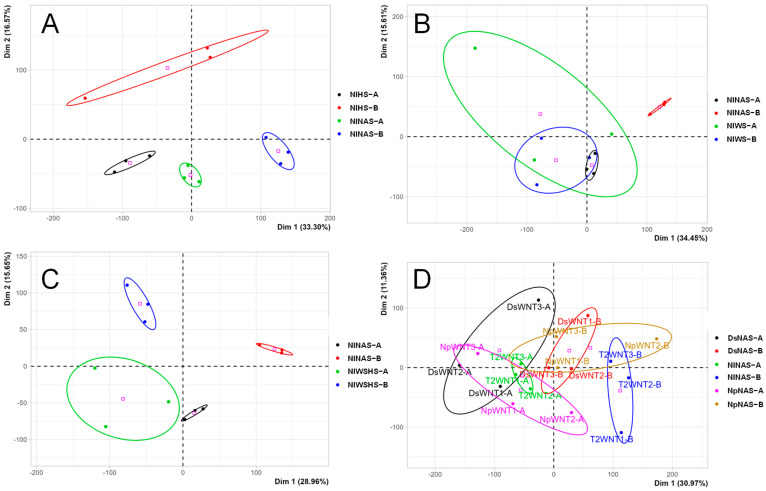
PCA analysis of transcriptomic data for a given category of stress (abiotic single, biotic, and abiotic combined). Only genes with a *p*-value < 0.05 (FDR-corrected) for a given category were considered in the PCA analysis. (**A**) Heat stress response. (**B**) Water stress response. (**C**) Fungal infection response. (**D**) Combined abiotic stress response.

**Figure 4 plants-12-00753-f004:**
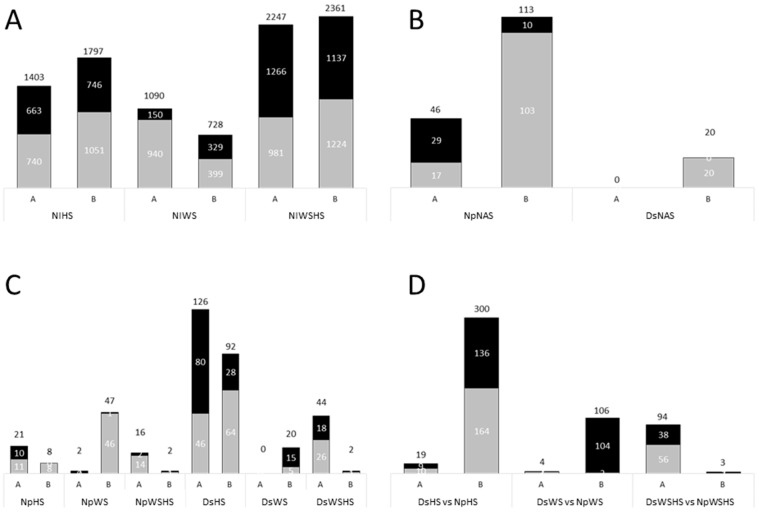
Number of differentially expressed genes. Gray: upregulated, black: downregulated. The number on top indicates the total DEGs. On the bottom of each graph, A denote timepoint A and B timepoint B. (**A**) Abiotic stress. (**B**) Single biotic stress. (**C**) Combination of abiotic and biotic stress. (**D**) Np- vs. Ds-infected samples.

**Figure 5 plants-12-00753-f005:**
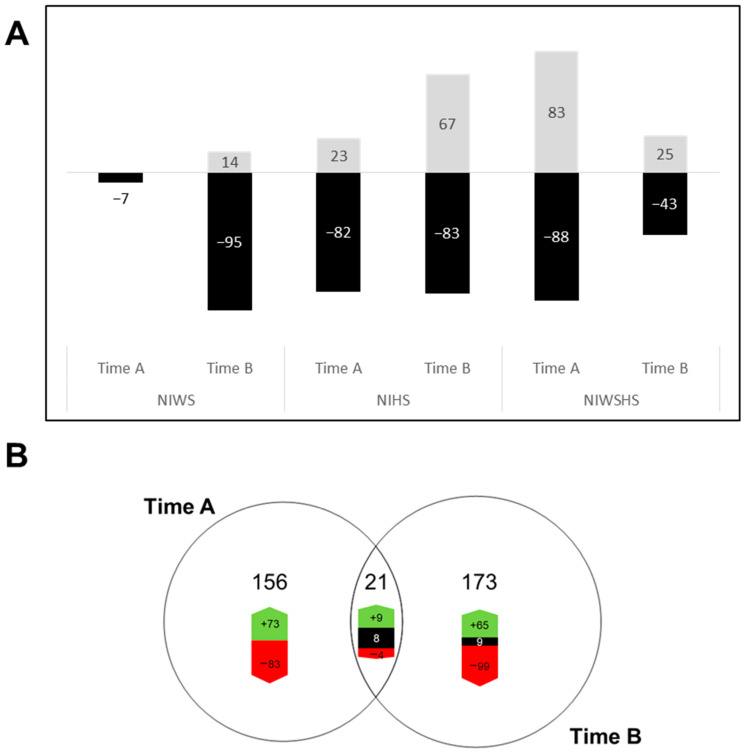
Effects of abiotic stresses, single or combined, on the leaf metabolome of noninfected vines. Non-inoculated plants (NI) were submitted to water stress (WS), heat stress (HS), or water stress and heat stress (WSHS), and their leaf metabolome was compared to non-stressed plants (NAS, no abiotic stress, as control). Time A: leaves were collected 8 days post the onset of water stress, 1 h post heat treatment, or 8 days post the onset of water stress and 1 h post heat treatment. Time B: leaves were collected 11 days post the onset of water stress, 3 days post heat treatment, or 11 days post the onset of water stress including 3 days of heat stress. (**A**) Regulation of significant *m*/*z* (T-test FDR < 0.05) determined from NI data subsets at timepoints A and B. The numbers of up- or down-accumulated *m*/*z* obtained for WS, HS, and WSHS conditions, compared to NAS, are indicated in the histogram bars (up in gray and down in black). (**B**) Venn diagram of the distribution of significantly regulated *m*/*z* at timepoints A and B of sampling. Green, black and red colors mean respectively either over-/none or under-accumulated metabolites.

**Figure 6 plants-12-00753-f006:**
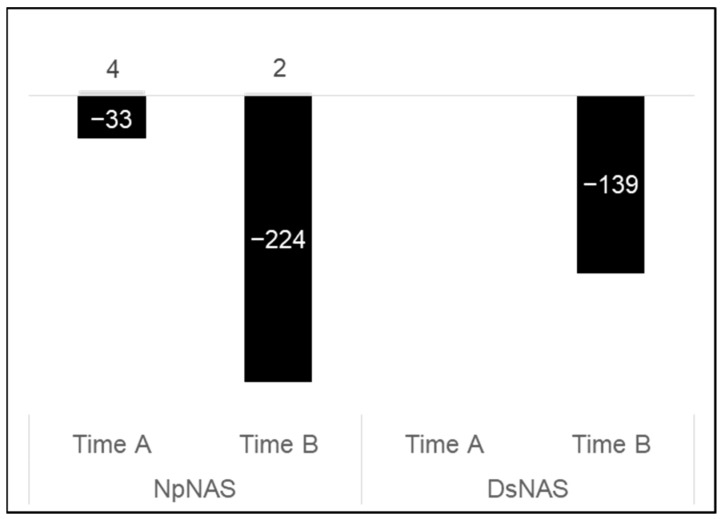
Effect of fungal infection on grapevine leaf metabolome. Vines grown in control conditions (NAS, no abiotic stress) were infected by *N. parvum* (Np) or *D. seriata* (Ds) at the level of a stem internode. Leaves collected above the area of infection were analyzed for metabolite profiling at 16 (timepoint A) and 19 (timepoint B) days post inoculation. Significant *m*/*z* (*t*-test FDR < 0.05) were determined from data subsets of timepoints A and B. Numbers of *m*/*z* up- (gray bars) or down-accumulated (black bars) compared to a non-inoculated control are indicated on histograms.

**Figure 7 plants-12-00753-f007:**
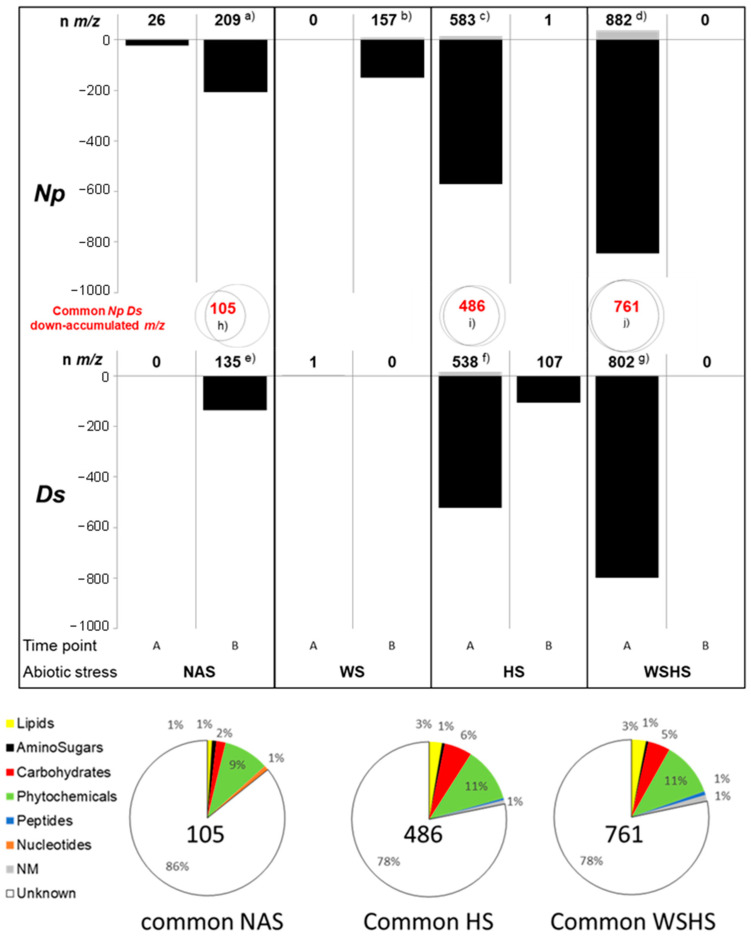
Effect of combined biotic and abiotic stresses on vine leaf metabolome. Vines were infected by *N. parvum* (Np) or *D. seriata* (Ds). They were either not stressed (NAS, no abiotic stress) or submitted to water stress (WS), heat stress (HS), or water stress and heat stress (WSHS), and their leaf metabolome was compared to the corresponding non-inoculated plants. Timepoint A: leaves were collected 8 days post the onset of water stress, 1 h post heat treatment, or 8 days post the onset of water stress and 1 h post heat treatment. Timepoint B: leaves were collected 11 days post the onset of water stress, 3 days post heat treatment, or 11 days post the onset of water stress including 3 days of heat stress. Significant *m*/*z* (*t*-test FDR < 0.05) were determined from data subsets of timepoints A and B. Total numbers of *m*/*z* (with validated raw formula), up- and down-accumulated compared to their respective control, are indicated on top of the histogram columns. Letters a to j refer to table lists of annotated compounds (Appendix A). Numbers of down-accumulated *m*/*z* common to Ds and Np samples are indicated in the Venn diagrams (**upper panel**) and pies (**bottom panel**), indicating the functional categories of common annotated compounds.

**Table 1 plants-12-00753-t001:** Results of ANOVAs performed with length of necrosis recorded on artificially infected plant. (**A**) Result of two-way ANOVA using infection with Np or Ds as factor 1 and combination of abiotic stress as factor 2. (**B**) Two-way ANOVA with only Np infected plant data (vs. control). (**C**) Two-way ANOVA with only Ds infected plant data (vs. control). *** and * indicate respectively a 0.1% and 5% significance value.

**A—All Necrosis Length Data Combined**	**Degree of Freedom**	**Sum of Square**	**Square Mean**	**F-Value**	***p*-Value**	**Significance**
Factor 1: infection	2	0.07289	0.03644	47.843	4.23 × 10^−9^	***
Faction 2: stress	3	0.00359	0.0012	1.572	0.222	
Interaction (infection × stress)	6	0.01235	0.00206	2.701	0.0379	*
Residuals	24	0.01828	0.00076			
**Coefficients:**		**Estimate**	**Std. Error**	***t* Value**	***p***-**Value**	**Significance**
(Intercept)		0.153963	0.015935	9.662	9.58 × 10^−10^	***
stressWS		−0.040229	0.022535	−1.785	0.0869	
stressHS		0.014863	0.022535	0.66	0.5158	
stressWSHS		0.016368	0.022535	0.726	0.4747	
infectionDs		−0.023705	0.022535	−1.052	0.3033	
infectionNp		−0.105748	0.022535	−4.693	9.07 × 10^−5^	***
stressWS:infectionDs		0.005667	0.031869	0.178	0.8604	
stressHS:infectionDs		−0.060717	0.031869	−1.905	0.0688	
stressWSHS:infectionDs		−0.088933	0.031869	−2.791	0.0101	*
stressWS:infectionNp		0.04796	0.031869	1.505	0.1454	
stressHS:infectionNp		−0.027043	0.031869	−0.849	0.4045	
stressWSHS:infectionNp		−0.038271	0.031869	−1.201	0.2415	
**B—Only Np Infected vs. Control Values**	**Degree of Freedom**	**Sum of Square**	**Square Mean**	**F-Value**	***p*-Value**	**Significance**
Factor 1: infection	1	0.07271	0.07271	91.219	5.19 × 10^−8^	***
Faction 2: stress	3	0.00117	0.00039	0.49	0.694	
Interaction (infection × stress)	3	0.00663	0.00221	2.773	0.0753	
Residuals	16	0.01275	0.0008			
**C—Only Np infected vs. Control Values**	**Degree of Freedom**	**Sum of Square**	**Square Mean**	**F-Value**	***p*-Value**	**Significance**
Factor 1: infection	1	0.021385	0.021385	24.586	0.000142	***
Faction 2: stress	3	0.004729	0.001576	1.812	0.185496	
Interaction (infection × stress)	3	0.009667	0.003222	3.705	0.033785	*
Residuals	16	0.013917	0.00087			

## Data Availability

Not applicable.

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
