# Peer review of "The Combination of Both Heat and Water Stresses May Worsen Botryosphaeria Dieback Symptoms in Grapevine"

_plants, 2023, doi:10.3390/plants12040753_

Round 1
Reviewer 1 Report
The paper titled “The combination of both heat and water stresses may worsen Botryosphaeria dieback symptoms in grapevine” investigates an important and original subject as it is one of the most important grapevine diseases.
The paper is well written and I recommend it for publication.
No corrections to add, just please avoid the abbreviations at the results and discussions specially when they are the first time appears and the paper needs very little English revision.
Overall, it is an excellent work
Author Response
Dear colleague,
We are very grateful for your remarks.
Because of the large number of conditions tested (twelve), we felt it was important to use abbreviations in order to facilitate the reading process. As you suggested, in the final editing process, we paid particular attention to the first occurrence of a given abbreviation in the article, providing the full meaning. In addition, we like to emphasize that the full list of abbreviations is listed in the material and methods section.
We hope this will satisfy you.
Kind regards
Olivier Fernandez
Reviewer 2 Report
The submitted article addresses grapevine trunk diseases, a current and extremely relevant problem, using a holistic methodological approach that combines fungal pathogenicity, abiotic factors and the transcriptomic and metabolomics effect assessment.
The article is well structured and reasoned. The methodologies and results obtained are clearly presented. Minor corrections are suggested to be consider:
The keyword list must not repeat words included in the title.
All scientific names of plant and fungal species must be written in italics.
Page 4- cold-response protein coding genes where down-regulated
6 - faster than HS stress alone
9 - Such fine tuning of irrigation may be m in the future
Legend of Figure 1 - all acronyms and abbreviations must be described in full
Legend of Figure 4 – (A) Abiotic stress
Supplementary figures S3, S4 and S5 are tables. The columns of the tables are cut off, making it impossible to read the information completely.
Author Response
Dear colleague,
We are very grateful for your comments. In the final revision process, we have taken into account all the minor points that you highlighted the best of our ability:
- Heat stress and drought stress, included in the title were replaced in the keywords list by abiotic stress and biotic stress.
- Scientific names, fungus and plant species were corrected to italic throughout the document
- In page 4 : “cold-response protein-coding genes where down-regulated” was corrected to “cold-response protein-coding genes were down-regulated”
- In page 6, I could not see the mistake
- In page 9 : “Such fine-tuning of irrigation may be m in the future” was corrected to “Such fine-tuning of irrigation may be more important in the future”
- Legends of figure 1 and 4 have been modified according to the reviewer’s request.
- We apologize for the poor quality of the initial tables/figures S3, S4 and S5 which was due to a bad copy/paste option choice. The tables have been updated. However, we hope that you will understand (since they are supplementary material) that, to avoid any errors in the final revised version, we have chosen to keep their original title (Supplementary figure).
We hope this will satisfy you.
Kind regards
Olivier Fernandez
Reviewer 3 Report
Well designed and performed study. I have no comments from the scientific point of view. Minor suggestions of formatting (species and generic names in italics, etc.) are highlighted directly within the manuscript attached.

Author Response
Dear colleague,
We thank you for your feedback. The italic formatting was lost in the copy/paste process and we have corrected it in the final revised version in addition to the other comments included in the pdf files.
We hope this will satisfy you.
Kind regards
Olivier Fernandez